# The association between vitamin D status and COVID-19 in England: A cohort study using UK Biobank

**Liang-Yu Lin** *, Amy Mulick, Rohini Mathur, Liam Smeeth, Charlotte Warren-Gash, Sinéad M. Langan**

Faculty of Epidemiology & Population Health, London School of Hygiene and Tropical Medicine, London, United Kingdom

* liang-yu.lin@lshtm.ac.uk

## Abstract

### Background

Recent studies indicate that vitamin D supplementation may decrease respiratory tract infections, but the association between vitamin D and COVID-19 is still unclear.

### Objective

To explore the association between vitamin D status and infections, hospitalisation, and mortality due to COVID-19.

### Methods

We used UK Biobank, a nationwide cohort of 500,000 individuals aged between 40 and 69 years at recruitment between 2006 and 2010. We included people with at least one serum vitamin D test, living in England with linked primary care and inpatient records. The primary exposure was serum vitamin D status measured at recruitment, defined as deficiency at <25 nmol/L, insufficiency at 25–49 nmol/L and sufficiency at ≥ 50 nmol/L. Secondary exposures were self-reported or prescribed vitamin D supplements. The primary outcome was laboratory-confirmed or clinically diagnosed SARS-CoV-2 infections. The secondary outcomes included hospitalisation and mortality due to COVID-19. We used multivariable Cox regression models stratified by summertime months and non-summertime months, adjusting for demographic factors and underlying comorbidities.

### Results

We included 307,512 participants (54.9% female, 55.9% over 70 years old) in our analysis. During summertime months, weak evidence existed that the vitamin D deficiency group had a lower hazard of being diagnosed with COVID-19 (hazard ratio [HR] = 0.86, 95% confidence interval [CI] = 0.77–0.95). During non-summertime, the vitamin D deficiency group had a higher hazard of COVID-19 compared with the vitamin D sufficient group (HR = 1.14,

**Data Availability Statement:** The database access dates were summarized in supporting information (S12 Table). The analytic codes of this project is shared on GitHub for review and re-use under MIT

open license (https://github.com/liang-yu12/vd_covid). Full pseudonymized participant data cannot be openly shared under the material transfer agreement with UK Biobank and ethics approval. Other researchers can apply for UK Biobank data to answer specific research questions. Further information about applying for data access can be obtained from the UK Biobank website (https://www.ukbiobank.ac.uk) or by emailing UK Biobank (ukbiobank@ukbiobank.ac.uk).

**Funding:** Liang-Yu Lin is funded by the scholarship of government sponsorship for overseas study by the Ministry of Education Taiwan. Rohini Mathur is funded by a Sir Henry Wellcome Postdoctoral Fellowship (201375/Z/16/Z). Charlotte Warren-Gash is supported by a Wellcome Intermediate Clinical Fellowship (201440/Z/16/Z). Sinéad Langan is funded by a Wellcome Senior Clinical Fellowship in Science (205039/Z/16/Z).

**Competing interests:** The authors have declared that no competing interests exist.

95% CI = 1.01–1.30). No evidence was found that vitamin D deficiency or insufficiency was associated with either hospitalisation or mortality due to COVID-19 in any time strata.

## Conclusion

We found no evidence of an association between historical vitamin D status and hospitalisation or mortality due to COVID-19, along with inconsistent results for any association between vitamin D and diagnosis of COVID-19. However, studies using more recent vitamin D measurements and systematic COVID-19 testing are needed.

## Introduction

The COVID-19 global pandemic is one of the biggest public health crises in recent history. The rapid spread of SARS-CoV-2 infection has caused serious casualties, overwhelming healthcare systems and disrupting societies. In the UK, more than 170,000 deaths due to COVID-19 within 28 days of a positive test were reported in the first year [1], planned surgeries and care have been delayed or cancelled [2], and prolonged lockdown measures along with the pandemic have worsened mental health [3]. Despite the introduction and distribution of COVID-19 vaccines by the end of 2020, controlling this pandemic at a global scale remains extremely difficult. Studying the aetiology of SARS-CoV-2 is important to inform effective prevention strategies in public health.

Vitamin D is essential to bone health for its ability in regulating calcium and phosphate homeostasis, and recent studies indicate it may have some immunomodulatory effects. At the cellular level, vitamin D can increase the production of antimicrobial peptides [4, 5] and regulates adaptive immunity response [6]. Clinically, a systematic review of observational studies indicated that vitamin D deficiency might be associated with longer duration of acute respiratory tract infection [7]. Another systematic review and meta-analysis including data from 37 original trials showed that vitamin D supplementation may protect against respiratory tract infections (pooled odds ratio = 0.92, 95% CI = 0.86–0.99) [8]. Because of its potential for preventing respiratory infections, vitamin D supplementation and fortification of food have been discussed as possible cheap public health interventions against COVID-19 [9].

Despite this potential, the association between vitamin D and COVID-19 is still unclear. If vitamin D deficiency is associated with COVID-19, vitamin D supplementation may be a potential public health intervention. Consequently, we aimed to conduct a historical cohort study using UK Biobank dataset and linked electronic health records, to better understand the association between serum vitamin D status, vitamin D supplementation and diagnosis of COVID-19 and outcomes.

## Methods

### Study population and eligibility

The study population was from UK Biobank, a nationwide cohort established between 2006 and 2010 [10]. In brief, participants aged 40 to 69 were recruited to 22 assessment centres around the UK. Their demographic information was collected through a touch screen questionnaire, and they received serum biochemical tests including vitamin D analysis. UK Biobank participants also gave their consent to have electronic health records linked, including primary care and inpatient care records, and death certificates [10, 11]. The primary care data

were provided by data system suppliers TPP and EMIS in England, and the inpatient care records and death records were provided by NHS digital. The external data providers extracted the health records by matching participant identifiers, including unique participant identifiers, NHS number, date of birth, sex and postcode. These health records were further processed and checked by UK Biobank before importing into the database [12]. We only included participants in England who had at least one serum vitamin D test, primary care registration records and inpatient care records. Those who lacked serum vitamin D test records, were not registered in England, did not have both inpatient and primary care registration records, were lost to follow-up, or died before 16 March 2020 were excluded. The distribution of demographic factors of the included and excluded participants were compared. The design of the cohort and inclusion and exclusion criteria are depicted in Fig 1.

## Primary exposure: Vitamin D status

The primary exposure was serum vitamin D status. The measurement of vitamin D levels in UK Biobank has been described previously [13]. In brief, serum vitamin D levels were measured when a participant visited a UK Biobank assessment centre, where their blood samples were collected and stored at -80˚C. Serum 25-hydroxyvitamin D status was measured using chemiluminescence immunoassay (DiaSorin Ltd. LIASON XL, Italy) in a centralised laboratory [14]. The testing process has been verified by quality control samples and through an external quality assurance scheme [15, 16]. Currently, no global consensus exists for determining vitamin D deficiency. We defined serum vitamin D status using Public Health England's definition (deficiency: <25 nmol/L; insufficiency: 25–50 nmol/L; sufficiency: $\geq$ 50 nmol/L)

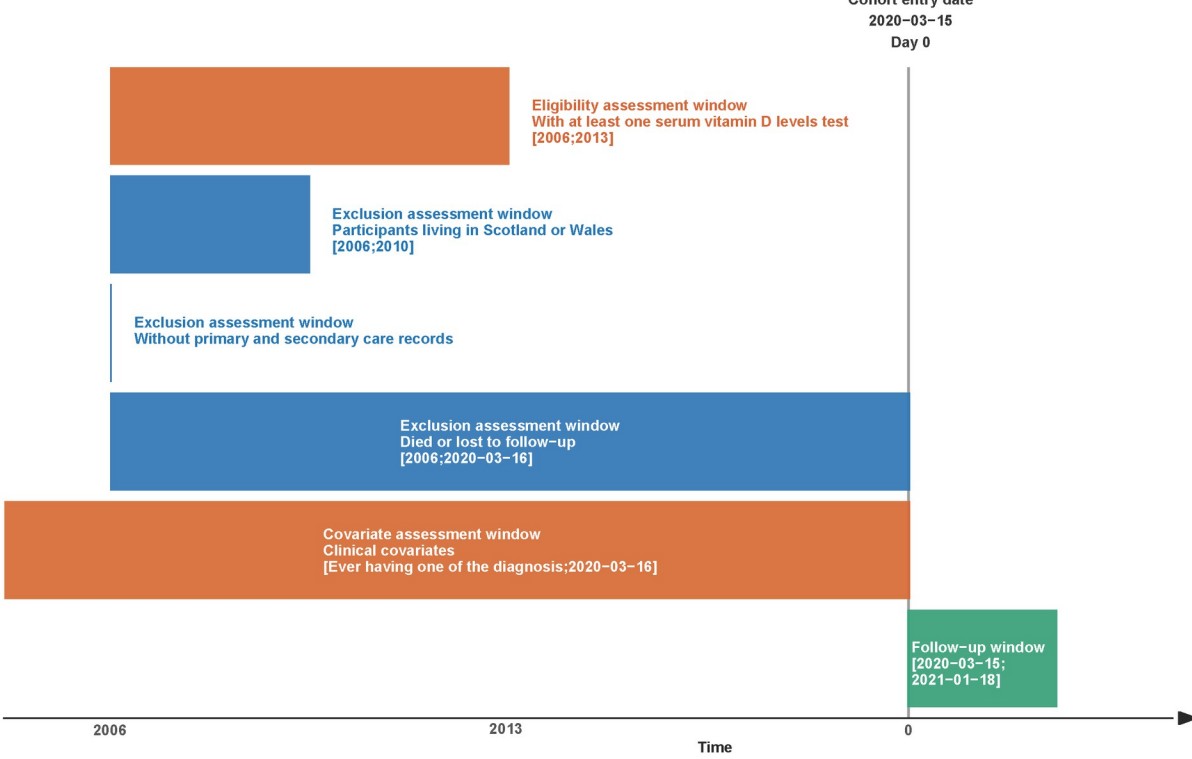

**Fig 1. Graphical depiction of the inclusion and exclusion criteria, cohort entry date and follow-up period.**

[17]. Participants who had their serum vitamin D levels tested between April and October were labelled as 'during summertime months,' and those who were had been tested between November and March were assigned as 'during non-summertime months.'

## Secondary exposure: Vitamin D supplementation and vitamin D prescription

The secondary exposures for this study were 1. taking vitamin D supplementation, or 2. receiving a vitamin D prescription from a GP. Information about vitamin D and other mineral supplementations was collected through a self-reported questionnaire using touch panels at the assessment centre between 2006 and 2010. We defined vitamin D supplementation as people who were taking vitamin D and associated minerals, including vitamin D, multivitamins, fish oil and calcium supplementation. Information about vitamin D supplementation was coded as 'taking vitamin D supplement' and 'not taking vitamin D supplement,' and it was coded as missing if a participant did not respond to the questionnaire.

Vitamin D prescriptions included all medications listed in British National Formula section 9.6.4, and we further compiled a prescription code list in Dictionary of Medicines and Devices (DM+D) using an existing mapping tool published by the NHS [18]. By using the DM+D code list, we identified participants who had ever received vitamin D prescriptions from the primary care prescription datasets. Vitamin D prescription was coded as 'had vitamin D prescriptions' and 'not receiving prescriptions.'

## Primary outcome: Diagnosis of COVID-19

The primary outcome of our study was the diagnosis of COVID-19, which was defined through laboratory testing or by clinical diagnosis of COVID-19. The laboratory tests for SARS-CoV-2 infection were performed using PCR, which was performed by the NHS (Pillar 1) or commercial partners (Pillar 2) [19, 20]. These testing results were reported to Public Health England and automatically imported into UK Biobank weekly [21]. Clinically diagnosed COVID-19 was defined as participants having diagnosis of COVID-19 codes in their electronic health records, either in primary care or inpatient care, or on the death certificate. We used existing code lists in CTV3 codes, SNOMED-CT and ICD-10 to identify the diagnosis of COVID-19.

## Secondary outcome ascertainment: Hospitalisation and mortality due to COVID-19

Hospitalisation due to COVID-19 was defined as COVID-19 related diagnosis (ICD-10 codes U071 or U072) recorded in the inpatient care dataset, and the admission date of each record was extracted. Mortality due to COVID-19 was defined as a participant having a COVID-19 diagnosis (ICD-10 codes U071 or U072) in the death registry data and being diagnosed as COVID-19 within 28 days, and the date of death was also recorded.

## Measurement of covariates

We included basic demographic factors associated with vitamin D deficiency or insufficiency in our model, described in our previous paper [13]. Demographic variables recorded between 2006 and 2010, such as sex, age, ethnicity, body mass index (BMI), alcohol drinking frequency, cigarette smoking, index of multiple deprivations (IMD), the time receiving serum vitamin D tests and the region of the UK Biobank assessment centre, were included in our analysis. The current age at the start of the pandemic was calculated from participants' year of birth, which

was coded as 'under 70 years old' and 'greater than and equal to 70 years old.' Other continuous covariates were further grouped into categorical variables. Self-reported ethnicity was classified as 'white,' 'black,' and 'Asian and others' according to the original questionnaire. BMI was grouped following National Institute for Health and Care Excellence guidelines for different sexes and ethnicities [22]. IMD scores were classified by five quintiles, and the quintile with the highest scores was assigned as 'most deprived.' We categorised the location of 22 UK Biobank assessment centres by the regions of England. Smoking statuses were coded as 'non-smoker', 'ex-smoker', or 'current-smoker' according to the original questionnaire. Regarding drinking frequency, participants were recoded as 'weekly' if participants reported drinking three or four times a week, and monthly if drinking one to three times a month was reported. Participants reported with 'prefer not to say' were labelled as missing value.

In addition, we included clinical covariates such as clinically extreme vulnerability and underlying chronic diseases. Participants who were clinically extremely vulnerable to COVID-19 were defined by Public Health England [23]. Underlying chronic diseases included hypertension, cardiovascular diseases, diabetes mellitus and asthma. Clinical covariates were assessed as a history of ever having one of the medical conditions of interest recorded in linked primary or secondary care records from the start of GP registration or HES recording until 16[th] March 2020. For health conditions such as chemoradiotherapy, blood cancer and bone marrow transplantation, we only included people who had a recent history in less than six months before the index date.

## Statistical analysis

The follow-up time of our study began on 15[th] March 2020. Because the availability of clinical datasets varied, the end of follow-up was defined differently for each outcome. For the primary outcome, SARS-CoV-2 infections, the event dates were the dates of diagnosis of COVID-19, and the censoring dates were the date of death or 18[th] January 2021. For hospitalisation due to COVID-19, the event dates were the dates of admission, and the censoring dates were the date of death or 30[th] November 2020. For mortality due to COVID-19, the event dates were the dates of death due to COVID-19, and the censoring dates were the dates of death due to other causes or 18[th] December 2020. In addition, among all UK Biobank participants with vitamin D testing data, we analysed the association between testing for vitamin D during the summertime months and vitamin D status using logistic regression.

The proportional hazard assumption was examined by using log(-log[survival]) plots. Due to the overlapping of survival curves, the assumptions about vitamin D status and the diagnosis of COVID-19 (S2a Fig), hospitalisation (S2b Fig), and mortality (S2c Fig) due to COVID-19 were violated. Therefore, we used stratified Cox regression to assess the association between vitamin D exposure and COVID-19 outcomes. The follow-up time of our models was stratified at 25[th] October 2020, which was the end date of the British summertime months. We carried out a crude analysis, then generated a partially adjusted model controlling sex and age, as well as a full model adjusting for all covariates. All statistical analysis was performed by using R Statistical Software (version 4.0.3, R Foundation for Statistical Computing, Vienna, Austria).

## Sensitivity analysis and model checking

The sensitivity analysis and justifications are summarised in Table 1. We repeated our analysis while changing the outcome to laboratory-confirmed SARS-CoV-2 infections, and we redid the analysis for hospitalisation and mortality among the subgroup with COVID-19 diagnosis. In addition, the association between receiving vitamin D tests during British summertime

**Table 1. Sensitivity analysis.**

| Sensitivity analysis | Justification |
|---|---|
| 1. The analysis was repeated on patients with laboratory-confirmed COVID-19. | The laboratory method is the gold standard for diagnosing COVID-19. |
| 2. The analysis of hospitalisation and mortality was repeated on a redefined cohort, which only included patients with laboratory-confirmed and clinically diagnosed COVID-19. | Because the strategies for testing COVID-19 have been changing over time, the COVID-19 diagnosis may be established in a different context. In the main analysis, we assessed the hospitalisation and mortality in the whole population at risk. To compare the severity of COVID-19 among patients with a confirmed diagnosis, we confined the analysis among subgroups with COVID-19 diagnosis, which had been made through clinical diagnosis or laboratory methods. |

months and vitamin D status was analysed using logistic regression adjusting for covariates among all participants with at least one vitamin D level test.

## Ethics

The UK Biobank data had been de-identified by removing personal data fields, such as postcodes and dates of birth [24]. The de-identified and anonymous data were released to eligible researchers for study purposes only. UK Biobank already has its Research Tissue Bank (RTB) approval from its Research Ethics Committee (REC), which covers most usage of the data under the UK Biobank ethics and governance framework [25]. The UK Biobank project was also approved by the Northwest Haydock Research Ethics Committee (reference: 11/NW/0382). Our project was approved by UK Biobank (ID:51265) and by the Research Ethics Committee of the London School of Hygiene and Tropical Medicine (reference: 17158). For COVID-19 data, an approved UK Biobank project will be automatically authorised to conduct COVID-19 related research after registering to access COVID-19 data [26]. We followed the principles of the Declaration of Helsinki [27].

## Results

### Study population

The selection of eligible participants is shown in Fig 2. After excluding ineligible people, a total of 307,512 participants were included in our analysis. The comparison of included and excluded participants is summarised in S1 Table. The distribution of sex, age, ethnicity, BMI, drinking behaviour, smoking, IMD, region and taking vitamin D supplementation was similar between included and excluded participants. More participants were clinically extremely vulnerable to COVID-19 or had underlying comorbidities among included participants with their electronic health records linked than those who were not eligible for inclusion.

Among eligible participants, more people were vitamin D sufficient (142,947; 46%) or insufficient (126,802; 41%) compared with vitamin D deficient participants (37,763, 12%). 65% received their vitamin D levels checked during British summertime months, while 35% were measured in non-summertime months. The distribution of demographic factors by vitamin D status is summarised in Table 2. The distribution of sex, taking vitamin D supplementation, region of residency, clinical vulnerability to COVID-19 and underlying chronic comorbidities was similar across different vitamin D groups. Compared to participants with insufficient or sufficient vitamin D levels, the vitamin D deficiency group had more participants who were under 70 years, non-white, obese and more deprived. Furthermore, the

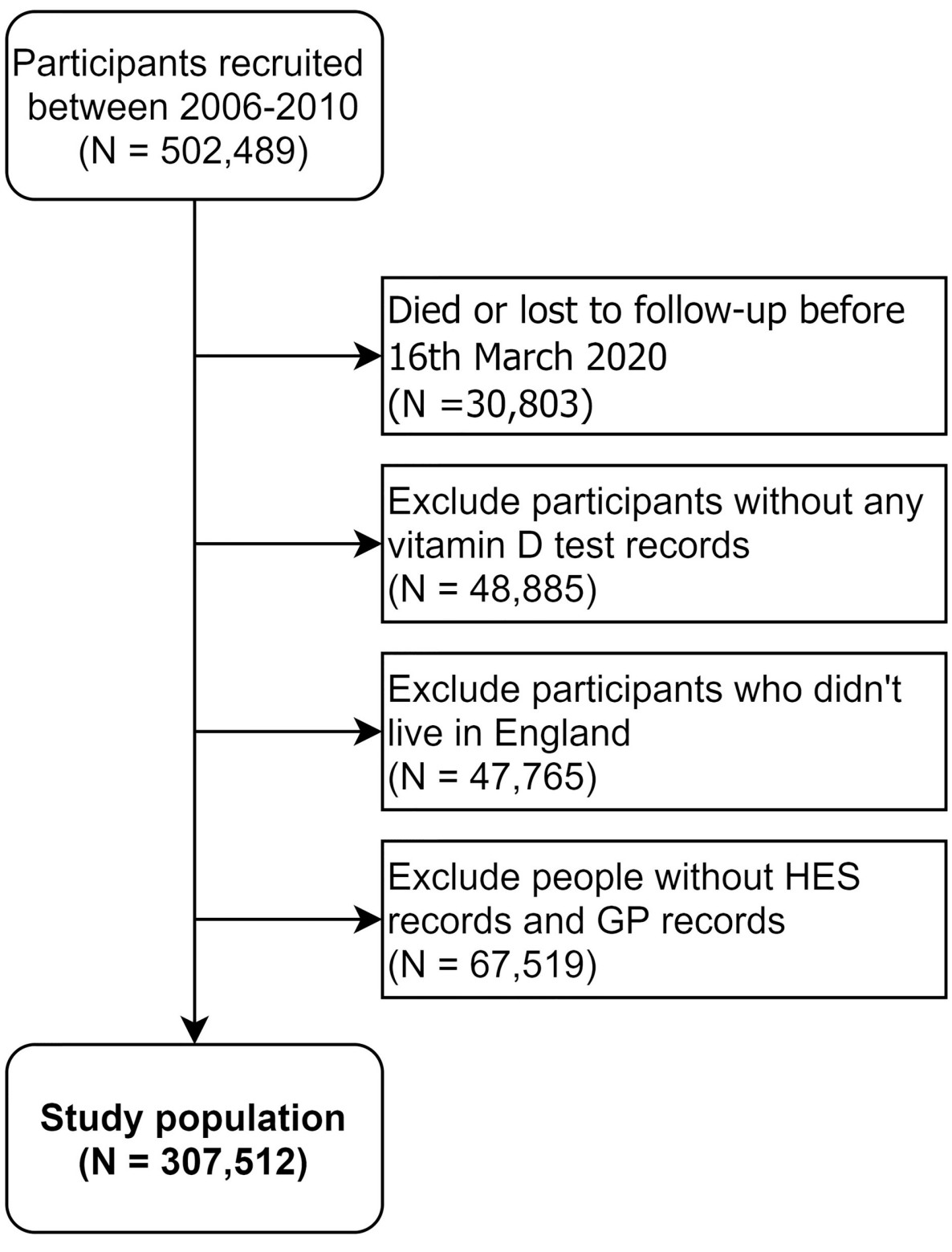

**Fig 2. Diagram of selecting study participants.**

**Table 2. The distribution of demographic characteristics by vitamin D status.**

| | Total (N = 307,512) | Vitamin D deficiency (<25 nmol/L) (N = 37,763) | Vitamin D insufficiency (25–50 nmol/L) (N = 126,802) | Vitamin D sufficiency (≥ 50 nmol/L) (N = 142,947) |
|---|---|---|---|---|
| The time when vitamin D sample was collected | | | | |
| • During non- summertime (November -March) | 108140 (35.2%) | 22312 (59.1%) | 51295 (40.5%) | 34533 (24.2%) |
| • During British summer time (April—October) | 199372 (64.8%) | 15451 (40.9%) | 75507 (59.5%) | 108414 (75.8%) |
| Sex | | | | |
| • Female | 169,018 (55.0%) | 20,706 (54.8%) | 69,860 (55.1%) | 78,452 (54.9%) |
| • Male | 138,494 (45.0%) | 17,057 (45.2%) | 56,942 (44.9%) | 64,495 (45.1%) |
| Age[1] | | | | |
| • Under 70 years old | 150,428 (48.9%) | 22,701 (60.1%) | 64,727 (51.0%) | 63,000 (44.1%) |
| • Greater and equal to 70 years old | 157,084 (51.1%) | 15,062 (39.9%) | 62,075 (49.0%) | 79,947 (55.9%) |
| Ethnicity | | | | |
| • White | 289,165 (94.0%) | 30,790 (81.5%) | 118,478 (93.4%) | 139,897 (97.9%) |
| • Black | 5,310 (1.7%) | 1,812 (4.8%) | 2,694 (2.1%) | 804 (0.6%) |
| • Asian and others | 13,037 (4.2%) | 5,161 (13.7%) | 5,630 (4.4%) | 2,246 (1.6%) |
| BMI[2] | | | | |
| • Healthy weight | 97,499 (31.8%) | 9,443 (25.2%) | 35,560 (28.2%) | 52,496 (36.8%) |
| • Underweight | 1,480 (0.5%) | 228 (0.6%) | 516 (0.4%) | 736 (0.5%) |
| • Overweight | 130,370 (42.6%) | 14,087 (37.6%) | 53,492 (42.4%) | 62,791 (44.0%) |
| • Obese | 76,989 (25.1%) | 13,661 (36.5%) | 36,732 (29.1%) | 26,596 (18.6%) |
| Drinking frequency | | | | |
| • Never | 24,394 (8.0%) | 5,504 (14.7%) | 10,492 (8.3%) | 8,398 (5.9%) |
| • Sometimes | 70,806 (23.1%) | 10,637 (28.3%) | 31,453 (24.9%) | 28,716 (20.1%) |
| • Weekly | 149,866 (48.8%) | 14,874 (39.6%) | 60,519 (47.8%) | 74,473 (52.2%) |
| • Daily | 61,770 (20.1%) | 6,533 (17.4%) | 24,054 (19.0%) | 31,183 (21.8%) |
| Drinking status | | | | |
| • Never | 13,434 (4.4%) | 3,487 (9.3%) | 5,806 (4.6%) | 4,141 (2.9%) |
| • Previous | 10,867 (3.5%) | 1,979 (5.3%) | 4,651 (3.7%) | 4,237 (3.0%) |
| • Current | 282,442 (92.1%) | 32,044 (85.4%) | 116,026 (91.7%) | 134,372 (94.1%) |
| Smoking status | | | | |
| • Non-smoker | 167,513 (54.8%) | 20,201 (54.0%) | 69,386 (55.0%) | 77,926 (54.8%) |
| • Ex-smoker | 108,326 (35.4%) | 11,315 (30.2%) | 43,916 (34.8%) | 53,095 (37.3%) |
| • Current-smoker | 30,105 (9.8%) | 5,926 (15.8%) | 12,875 (10.2%) | 11,304 (7.9%) |
| IMD[3] | | | | |
| • Least deprived | 59,870 (20.0%) | 4,945 (13.5%) | 23,069 (18.7%) | 31,856 (22.9%) |
| • 2 deprived | 59,219 (19.8%) | 5,525 (15.1%) | 23,876 (19.3%) | 29,818 (21.4%) |
| • 3 deprived | 60,261 (20.1%) | 6,551 (17.9%) | 24,501 (19.9%) | 29,209 (21.0%) |
| • 4 deprived | 60,255 (20.1%) | 8,313 (22.7%) | 25,560 (20.7%) | 26,382 (19.0%) |
| • Most deprived | 59,490 (19.9%) | 11,230 (30.7%) | 26,414 (21.4%) | 21,846 (15.7%) |
| Vitamin D and mineral supplementation[4] | | | | |
| • Not taking supplement | 22417 (21.2%) | 2791 (32.6%) | 9252 (24.0%) | 10374 (17.8%) |
| • Taking vitamin D supplement | 83131 (78.8%) | 5773 (67.4%) | 29364 (76.0%) | 47994 (82.2%) |
| Vitamin D prescription[5] | | | | |
| • Not receiving prescriptions | 235431 (76.6%) | 25567 (67.7%) | 97094 (76.6%) | 112770 (78.9%) |
| • Had vitamin D prescriptions | 72081 (23.4%) | 12196 (32.3%) | 29708 (23.4%) | 30177 (21.1%) |

(*Continued*)

**Table 2.** (Continued)

| | Total (N = 307,512) | Vitamin D deficiency (<25 nmol/L) (N = 37,763) | Vitamin D insufficiency (25–50 nmol/L) (N = 126,802) | Vitamin D sufficiency (≥ 50 nmol/L) (N = 142,947) |
|---|---|---|---|---|
| Region of UK Biobank assessment centres | | | | |
| • East Midlands | 43707 (14.2%) | 5,609 (14.9%) | 17,643 (13.9%) | 20,455 (14.3%) |
| • London | 24467 (8.0%) | 2,350 (6.2%) | 9,711 (7.7%) | 12,406 (8.7%) |
| • North East | 44374 (14.4%) | 7,446 (19.7%) | 19,774 (15.6%) | 17,154 (12.0%) |
| • North West | 50808 (16.5%) | 5,916 (15.7%) | 20,569 (16.2%) | 24,323 (17.0%) |
| • South East | 28859 (9.4%) | 2,329 (6.2%) | 11,192 (8.8%) | 15,338 (10.7%) |
| • South West | 29445 (9.6%) | 2,271 (6.0%) | 11,103 (8.8%) | 16,071 (11.2%) |
| • West Midlands | 31522 (10.3%) | 5,249 (13.9%) | 13,919 (11.0%) | 12,354 (8.6%) |
| • Yorkshire and The Humber | 54330 (17.7%) | 6,593 (17.5%) | 22,891 (18.1%) | 24,846 (17.4%) |
| Clinically vulnerable to COVID-19[6,7] | | | | |
| • Not vulnerable | 249,944 (81.3%) | 29,903 (79.2%) | 103,063 (81.3%) | 116,978 (81.8%) |
| • Clinically extremely vulnerable | 57,568 (18.7%) | 7,860 (20.8%) | 23,739 (18.7%) | 25,969 (18.2%) |
| Other comorbidities[6,8] | | | | |
| • No chronic diseases | 94,237 (30.6%) | 10,466 (27.7%) | 38,186 (30.1%) | 45,585 (31.9%) |
| • With Chronic diseases | 213,275 (69.4%) | 27,297 (72.3%) | 88,616 (69.9%) | 97,362 (68.1%) |

[1]. Calculated from participants' year of birth.

[2]. The classification is suggested by NICE guidelines.

[3]. IMD scores were classified by quintile.

[4]. Vitamin D supplement includes vitamin D, multivitamin, fish oil and calcium supplementation.

[5]. Vitamin D prescription included all drugs in BNF section 9.6.4, which were identified by using code lists in DM+D codes from linked GP prescription records.

[6]. Health conditions were identified from linked electronic health records.

[7]. The clinically extremely vulnerable groups were defined by using Public Health England's definition.

[8]. Including hypertension, cardiovascular diseases, diabetes mellitus, or asthma.

proportions of alcohol drinking and taking vitamin D supplements were lower among the vitamin D deficiency group.

## Description of outcomes

The distribution of diagnosis of COVID-19, hospitalisation, and mortality due to COVID-19 over time is summarised in S1 Fig. As can be seen in S1a Fig, among 10,165 participants with SARS-CoV-2 infection, more participants were diagnosed with COVID-19 in spring (13.8%), autumn (51.4%), and winter (31%), while fewer cases were reported in summer (3.8%). Despite the shorter follow-up period, similar distributions were also noted for hospitalisation (S1b Fig) and mortality (S1c Fig). In the larger cohort containing all participants with vitamin D records, we found that participants visiting the UK Biobank assessment centre during British summertime months had around 60% lower odds of vitamin D deficiency or insufficiency than those receiving tests during non-summertime months (S2 Table).

## Association between vitamin D status and diagnosis of COVID-19

Table 3 summarises the association between vitamin D status and being diagnosed with COVID-19. In crude analysis, in British summertime months, people with vitamin D insufficiency or deficiency had a higher hazard of being diagnosed with COVID-19 than sufficient participants (insufficiency: crude HR = 1.18, CI = 1.07–1.31; deficiency: crude HR = 1.11,

**Table 3. The association between vitamin D status and diagnosis of COVID-19.**

| | | HR (95% CI, p-value) (crude) | HR (95% CI, p-value) (adjusted for sex and age) | HR (95% CI, p-value) (adjusted for all covariates) |
|---|---|---|---|---|
| **British summertime** (15 March to 25 October 2020) | Vitamin D sufficiency | Reference | Reference | Reference |
| | Vitamin D insufficiency | 1.18 (1.07–1.31, p<0.01) | 1.07 (1.00–1.15, p = 0.06) | 0.96 (0.90–1.04, p = 0.32) |
| | Vitamin D deficiency | 1.11 (1.03–1.19, p<0.01) | 1.08 (0.98–1.20, p = 0.12) | 0.86 (0.77–0.95, p<0.01) |
| **Non-summertime** (26 October to 18 January 2021) | Vitamin D sufficiency | Reference | Reference | Reference |
| | Vitamin D insufficiency | 0.93 (0.86–1.02, p = 0.12) | 0.93 (0.85–1.02, p = 0.12) | 0.93 (0.85–1.02, p = 0.13) |
| | Vitamin D deficiency | 1.15 (1.02–1.31, p = 0.02) | 1.15 (1.02–1.31, p = 0.02) | 1.14 (1.01–1.30, p = 0.04) |
| Sex | Female | Reference | - | Reference |
| | Male | 1.10 (1.06–1.15, p<0.01) | - | 1.08 (1.04–1.13, p<0.01) |
| Age[1] | Under 70 years old | Reference | - | Reference |
| | Greater and equal to 70 years old | 0.59 (0.56–0.61, p<0.01) | - | 0.57 (0.54–0.59, p<0.01) |
| Ethnicity | White | Reference | Reference | Reference |
| | Black | 1.79 (1.60–2.01, p<0.01) | 1.58 (1.41–1.78, p<0.01) | 1.36 (1.20–1.53, p<0.01) |
| | Asian and others | 1.72 (1.60–1.86, p<0.01) | 1.57 (1.45–1.70, p<0.01) | 1.43 (1.31–1.56, p<0.01) |
| BMI[2] | Healthy weight | Reference | Reference | Reference |
| | Underweight | 1.10 (0.81–1.49, p = 0.53) | 1.08 (0.80–1.47, p = 0.6) | 1.06 (0.78–1.44, p = 0.71) |
| | Overweight | 1.23 (1.17–1.29, p<0.01) | 1.26 (1.19–1.32, p<0.01) | 1.20 (1.14–1.26, p<0.01) |
| | Obese | 1.60 (1.52–1.69, p<0.01) | 1.62 (1.54–1.71, p<0.01) | 1.44 (1.36–1.52, p<0.01) |
| Drinking frequency | Never | Reference | Reference | Reference |
| | Sometimes | 0.87 (0.81–0.94, p<0.01) | 0.86 (0.80–0.92, p<0.01) | 0.94 (0.87–1.01, p = 0.11) |
| | Weekly | 0.80 (0.74–0.85, p<0.01) | 0.77 (0.72–0.82, p<0.01) | 0.93 (0.86–1.00, p = 0.05) |
| | Daily | 0.65 (0.60–0.70, p<0.01) | 0.64 (0.60–0.70, p<0.01) | 0.81 (0.75–0.89, p<0.01) |
| Smoking status | Non-smoker | Reference | Reference | Reference |
| | Ex-smoker | 1.09 (1.04–1.14, p<0.01) | 1.16 (1.11–1.21, p<0.01) | 1.15 (1.10–1.20, p<0.01) |
| | Current smoker | 1.23 (1.15–1.31, p<0.01) | 1.15 (1.08–1.23, p<0.01) | 1.06 (0.99–1.13, p = 0.11) |
| Vitamin D status testing time | During non-summertime | Reference | Reference | Reference |
| | During British summer time | 0.98 (0.94–1.02, p = 0.32) | 0.99 (0.95–1.03, p = 0.49) | 1.00 (0.96–1.04, p = 0.93) |
| IMD[3] | Least deprived | Reference | Reference | Reference |
| | 2 deprived | 1.18 (1.10–1.27, p<0.01) | 1.18 (1.10–1.27, p<0.01) | 1.09 (1.02–1.18, p = 0.02) |
| | 3 deprived | 1.38 (1.28–1.48, p<0.01) | 1.36 (1.27–1.46, p<0.01) | 1.21 (1.13–1.30, p<0.01) |
| | 4 deprived | 1.63 (1.53–1.75, p<0.01) | 1.59 (1.49–1.70, p<0.01) | 1.35 (1.26–1.45, p<0.01) |
| | Most deprived | 2.16 (2.03–2.30, p<0.01) | 2.05 (1.92–2.19, p<0.01) | 1.59 (1.48–1.70, p<0.01) |

*(Continued)*

**Table 3.** (Continued)

| | | HR (95% CI, p-value) (crude) | HR (95% CI, p-value) (adjusted for sex and age) | HR (95% CI, p-value) (adjusted for all covariates) |
|---|---|---|---|---|
| Regions | North East | Reference | Reference | Reference |
| | East Midlands | 0.87 (0.80–0.95, p<0.01) | 0.88 (0.80–0.96, p<0.01) | 0.92 (0.84–1.01, p = 0.07) |
| | London | 1.04 (0.97–1.12, p = 0.29) | 1.01 (0.94–1.09, p = 0.75) | 0.92 (0.85–0.99, p = 0.03) |
| | North West | 1.30 (1.22–1.39, p<0.01) | 1.31 (1.22–1.39, p<0.01) | 1.24 (1.16–1.32, p<0.01) |
| | South East | 0.57 (0.52–0.63, p<0.01) | 0.57 (0.52–0.63, p<0.01) | 0.66 (0.60–0.73, p<0.01) |
| | South West | 0.66 (0.60–0.73, p<0.01) | 0.65 (0.59–0.71, p<0.01) | 0.70 (0.64–0.77, p<0.01) |
| | West Midlands | 1.03 (0.96–1.12, p = 0.41) | 1.02 (0.94–1.10, p = 0.66) | 0.95 (0.88–1.03, p = 0.24) |
| | Yorkshire and The Humber | 0.97 (0.91–1.04, p = 0.38) | 0.96 (0.90–1.03, p = 0.27) | 0.96 (0.90–1.03, p = 0.28) |
| Clinically vulnerable to COVID-19[4] | Not vulnerable | Reference | Reference | Reference |
| | Extremely vulnerable | 1.28 (1.23–1.35, p<0.01) | 1.42 (1.36–1.49, p<0.01) | 1.29 (1.23–1.35, p<0.01) |
| Underlying comorbidities[5] | No chronic diseases | Reference | Reference | Reference |
| | Chronic diseases | 1.07 (1.02–1.12, p<0.01) | 1.21 (1.15–1.26, p<0.01) | 1.02 (0.97–1.07, p = 0.43) |

[1]. Calculated from participants' year of birth.

[2]. The classification is suggested by NICE guidelines.

[3]. IMD scores were classified by quintile.

[4]. The clinically extremely vulnerable groups were defined by using Public Health England's definition.

[5]. Including hypertension, cardiovascular diseases, diabetes mellitus, and asthma

CI = 1.03–1.19), but in non-summertime months, only the vitamin D insufficiency group had a higher hazard of COVID-19 (insufficiency: crude HR = 1.15, CI = 1.02–1.31.) The results of the partially adjusted model were similar: only vitamin D deficiency in non-summertime was associated with an increased hazard of COVID-19 (HR = 1.15, CI = 1.02–1.31). After adjusting for all covariates, participants with vitamin D deficiency had a 14% lower hazard of being diagnosed with COVID-19 compared with people with sufficient vitamin D status in British summertime months (HR = 0.86, CI = 0.77–0.95). In non-summertime months, the hazard of being diagnosed with COVID-19 was 14% higher among the vitamin D deficiency group (HR = 1.14, CI = 1.01–1.30). No evidence showed that vitamin D insufficiency was associated with COVID-19 during either summertime or non-summertime months. Being male, non-white, overweight or obese, more deprived, clinically vulnerable or with underlying comorbidities was associated with an increased hazard of being diagnosed with COVID-19, while people who were older than 70 years had a lower hazard of being diagnosed (Table 3).

## Association between vitamin D status and hospitalisation due to COVID-19

Table 4 summarises the association between serum vitamin D status and hospitalisation due to COVID-19 stratified by summertime months. In the crude and partially adjusted models, in British summertime months, vitamin D insufficiency or deficiency was associated with a

**Table 4. The association between vitamin D status and hospitalisation due to COVID-19.**

| | | HR (crude) | HR (adjusted for sex and age) | HR (adjusted for all covariates) |
|---|---|---|---|---|
| **British summertime** (15 March to 25 October 2020) | Vitamin D sufficiency | Reference | Reference | Reference |
| | Vitamin D insufficiency | 1.13 (0.99–1.29, p = 0.07) | 1.18 (1.03–1.34, p = 0.01) | 0.94 (0.82–1.08, p = 0.38) |
| | Vitamin D deficiency | 1.50 (1.26–1.79, p<0.01) | 1.66 (1.40–1.98, p<0.01) | 1.08 (0.89–1.31, p = 0.42) |
| **Non-summertime** (26 October to 30 November 2020) | Vitamin D sufficiency | Reference | Reference | Reference |
| | Vitamin D insufficiency | 1.05 (0.79–1.39, p = 0.73) | 1.05 (0.79–1.39, p = 0.73) | 1.11 (0.83–1.49, p = 0.46) |
| | Vitamin D deficiency | 0.92 (0.63–1.35, p = 0.67) | 0.92 (0.63–1.35, p = 0.67) | 0.92 (0.61–1.37, p = 0.67) |
| Sex | Female | Reference | - | Reference |
| | Male | 1.96 (1.76–2.18, p<0.01) | - | 1.72 (1.53–1.93, p<0.001) |
| Age[1] | Under 70 years old | Reference | - | Reference |
| | Greater and equal to 70 years old | 1.80 (1.61–2.01, p<0.01) | - | 1.50 (1.32–1.69, p<0.001) |
| Ethnicity | White | Reference | Reference | Reference |
| | Black | 2.22 (1.67–2.95, p<0.01) | 2.75 (2.07–3.66, p<0.01) | 2.17 (1.59–2.94, p<0.001) |
| | Asian and others | 1.59 (1.28–1.97, p<0.01) | 1.77 (1.43–2.20, p<0.01) | 1.39 (1.08–1.79, p = 0.012) |
| BMI[2] | Healthy weight | Reference | Reference | Reference |
| | Underweight | 2.00 (0.94–4.23, p = 0.07) | 2.31 (1.09–4.89, p = 0.03) | 1.97 (0.93–4.19, p = 0.078) |
| | Overweight | 1.80 (1.55–2.10, p<0.01) | 1.56 (1.33–1.82, p<0.01) | 1.43 (1.21–1.68, p<0.001) |
| | Obese | 3.05 (2.61–3.55, p<0.01) | 2.76 (2.37–3.22, p<0.01) | 2.05 (1.74–2.42, p<0.001) |
| Drinking frequency | Never | Reference | Reference | Reference |
| | Sometimes | 0.67 (0.56–0.80, p<0.01) | 0.70 (0.58–0.83, p<0.01) | 0.79 (0.65–0.96, p = 0.015) |
| | Weekly | 0.53 (0.45–0.63, p<0.01) | 0.48 (0.41–0.57, p<0.01) | 0.68 (0.57–0.82, p<0.001) |
| | Daily | 0.53 (0.43–0.64, p<0.01) | 0.43 (0.35–0.52, p<0.01) | 0.63 (0.51–0.78, p<0.001) |
| Smoking status | Non-smoker | Reference | Reference | Reference |
| | Ex-smoker | 1.61 (1.44–1.81, p<0.01) | 1.41 (1.26–1.59, p<0.01) | 1.29 (1.14–1.46, p<0.001) |
| | Current smoker | 1.94 (1.65–2.28, p<0.01) | 1.88 (1.60–2.22, p<0.01) | 1.42 (1.19–1.69, p<0.001) |
| Vitamin D status testing time | During non- summertime | Reference | Reference | Reference |
| | During British summer time | 0.99 (0.89–1.11, p = 0.91) | 1.01 (0.90–1.13, p = 0.87) | 1.04 (0.92–1.17, p = 0.519) |
| IMD[3] | Least deprived | Reference | Reference | Reference |
| | 2 deprived | 1.19 (0.96–1.46, p = 0.11) | 1.19 (0.96–1.47, p = 0.11) | 1.06 (0.85–1.31, p = 0.609) |
| | 3 deprived | 1.37 (1.12–1.68, p<0.01) | 1.39 (1.13–1.70, p<0.01) | 1.12 (0.90–1.37, p = 0.307) |
| | 4 deprived | 1.82 (1.50–2.20, p<0.01) | 1.88 (1.55–2.28, p<0.01) | 1.41 (1.15–1.72, p = 0.001) |
| | Most deprived | 2.87 (2.39–3.43, p<0.01) | 3.04 (2.54–3.64, p<0.01) | 1.78 (1.46–2.16, p<0.001) |

(*Continued*)

**Table 4.** (Continued)

| | | HR (crude) | HR (adjusted for sex and age) | HR (adjusted for all covariates) |
|---|---|---|---|---|
| Regions | North East | Reference | Reference | Reference |
| | East Midlands | 1.18 (0.94–1.48, p = 0.16) | 1.17 (0.93–1.47, p = 0.19) | 1.30 (1.03–1.65, p = 0.028) |
| | London | 0.88 (0.71–1.09, p = 0.24) | 0.92 (0.74–1.13, p = 0.42) | 0.79 (0.63–0.99, p = 0.044) |
| | North West | 1.54 (1.28–1.84, p<0.01) | 1.52 (1.27–1.83, p<0.01) | 1.39 (1.16–1.68, p<0.001) |
| | South East | 0.49 (0.37–0.66, p<0.01) | 0.49 (0.37–0.66, p<0.01) | 0.67 (0.49–0.90, p = 0.009) |
| | South West | 0.58 (0.44–0.76, p<0.01) | 0.60 (0.45–0.78, p<0.01) | 0.69 (0.51–0.91, p = 0.010) |
| | West Midlands | 1.34 (1.09–1.65, p<0.01) | 1.33 (1.08–1.64, p<0.01) | 1.21 (0.97–1.50, p = 0.084) |
| | Yorkshire and The Humber | 1.20 (1.00–1.45, p = 0.05) | 1.21 (1.00–1.46, p = 0.05) | 1.27 (1.05–1.55, p = 0.014) |
| Clinically vulnerable to COVID-19[4] | Not vulnerable | Reference | Reference | Reference |
| | Extremely vulnerable | 3.50 (3.14–3.89, p<0.01) | 3.20 (2.87–3.57, p<0.01) | 2.55 (2.28–2.86, p<0.001) |
| Underlying comorbidities[5] | No chronic diseases | Reference | Reference | Reference |
| | Chronic diseases | 2.84 (2.43–3.31, p<0.01) | 2.41 (2.06–2.82, p<0.01) | 1.61 (1.36–1.90, p<0.001) |

[1]. Calculated from participants' year of birth.

[2]. The classification is suggested by NICE guidelines.

[3]. IMD scores were classified by quintile.

[4]. The clinically extremely vulnerable groups were defined by using Public Health England's definition.

[5]. Including hypertension, cardiovascular diseases, diabetes mellitus, and asthma

higher hazard of hospitalisation due to COVID-19, while in non-summertime, such an association was not seen. We found either during or after British summertime months, after adjusting for covariates and compared with people with sufficient vitamin D status, no evidence existed that vitamin D insufficiency or deficiency was associated with a higher hazard of hospital admission due to COVID-19 (during British summertime months: insufficiency adjusted HR = 0.94, CI = 0.82–1.08, deficiency adjusted HR = 1.08, CI = 0.89–1.31; during non-summertime: insufficiency adjusted HR = 1.11, CI = 0.83–1.49, deficiency adjusted HR = 0.92, CI = 0.61–1.37). Other covariates such as male sex, age older than 70 years, non-white ethnicity, overweight or obesity, cigarette smoking, and being more deprived, clinically vulnerable or having underlying comorbidities increased the hazard of hospitalisation due to COVID-19. Compared with participants who never drink alcohol, more frequent alcohol drinking was associated with a decreased hazard of hospitalisation (Table 4).

## Association between vitamin D status and COVID-19 mortality

Table 5 summarises the association between vitamin D status and mortality due to COVID-19. In the crude and partially adjusted model, no association was found between vitamin D status and COVID-19 mortality, except vitamin D deficiency during summertime months had higher risk of dying from COVID-19, after adjusting for sex and age (partially adjusted HR = 1.64, CI = 1.06–2.54.) Compared with people with sufficient vitamin D status and after

adjusting for covariates, no evidence existed that the hazard of COVID-19 mortality was higher among participants with vitamin D insufficiency or deficiency, either during or after British summertime months (during British summertime months: insufficiency adjusted HR = 0.84, CI = 0.60–1.17, deficiency adjusted HR = 1.08, CI = 0.68–1.72; during non-summertime: insufficiency adjusted HR = 1.35, CI = 0.79–2.30, deficiency adjusted HR = 1.46, CI = 0.73–2.91). In addition, male sex, age over 70 years, black ethnicity, underweight and obesity, cigarette smoking, being most deprived, clinical vulnerability and having underlying comorbidities were associated with an increased hazard of COVID-19 mortality. Frequent alcohol drinking was associated with a decreased hazard of COVID-19 mortality (Table 5).

### Association between vitamin D prescription or supplementation and COVID-19

Some evidence existed that during summertime months, people who had been ever prescribed vitamin D supplementation from a GP had a higher hazard of being diagnosed with COVID-19 (S3 Table, adjusted HR = 1.22, CI = 1.13–1.32), hospitalisation (S4 Table, adjusted HR = 1.59, CI = 1.39–1.82) and mortality (S5 Table, adjusted HR = 2.31, CI = 1.68–3.17). During British summertime months, no evidence showed self-reported vitamin D supplementation was associated with a lower hazard of diagnosis of COVID-19 (adjusted HR = 0.88, CI = 0.76–1.01), while the hazard was higher during non-summertime (S6 Table, adjusted HR = 1.23, CI = 1.03–1.47). No evidence was found that self-reported vitamin D supplementation was associated with hospitalisation (S7 Table) or mortality (S8 Table) due to COVID-19 either during or after summertime months.

### Sensitivity analysis

The repeated analysis of vitamin D status and laboratory-confirmed COVID-19 was similar to the original model (S9 Table). Similarly, the subgroup analysis of hospitalisation and mortality among patients with diagnosis of COVID-19 showed that there was no evidence that vitamin D status was associated with the hazard of COVID-19 hospitalisation or mortality (S10 and S11 Tables).

## Discussion

In this large cohort study, we found no consistent evidence that historical vitamin D status was associated with COVID-19. No evidence showed that historical evidence of vitamin D deficiency or insufficiency was associated with hospitalisation or mortality due to COVID-19. During British summertime months, weak evidence existed that vitamin D deficiency was associated with a lower hazard of being diagnosed with COVID-19, while during non-summertime, the association was reversed. In the secondary analysis, during summertime months, people who ever received vitamin D prescription had a higher hazard of having diagnosis of COVID-19, hospitalisation, and mortality due to COVID-19. No association was found between self-reported vitamin D supplementation and hospitalisation or mortality due to COVID-19.

Our study has some strengths. First, compared to previous studies using UK Biobank datasets early in the pandemic, the follow-up period of our study was longer, and therefore we were able to cover more than one wave of COVID-19 infections [28–31]. Second, our analysis adjusted for more clinical covariates using the latest electronic health records, allowing us to estimate the effect of vitamin D status more accurately. Third, despite the variation of COVID-19 testing strategies, the clinical outcomes of hospitalisation and mortality were collected in a systematic way, which minimised the misclassification bias of these outcomes. The large

**Table 5. The association between vitamin D status and COVID-19 mortality.**

| | | HR (crude) | HR (adjusted for sex and age) | HR (adjusted for all covariates) |
|---|---|---|---|---|
| **British summertime** (15 March to 25 October 2020) | Vitamin D sufficiency | Reference | Reference | Reference |
| | Vitamin D insufficiency | 0.95 (0.69–1.31, p = 0.74) | 1.06 (0.77–1.46, p = 0.72) | 0.84 (0.60–1.17, p = 0.30) |
| | Vitamin D deficiency | 1.26 (0.82–1.95, p = 0.29) | 1.64 (1.06–2.54, p = 0.03) | 1.08 (0.68–1.72, p = 0.75) |
| **Non-summertime** (26 October to 18 December 2020) | Vitamin D sufficiency | Reference | Reference | Reference |
| | Vitamin D insufficiency | 1.22 (0.72–2.05, p = 0.46) | 1.22 (0.72–2.05, p = 0.46) | 1.35 (0.79–2.30, p = 0.28) |
| | Vitamin D deficiency | 1.34 (0.67–2.65, p = 0.41) | 1.34 (0.68–2.65, p = 0.40) | 1.46 (0.73–2.91, p = 0.29) |
| Sex | Female | Reference | - | Reference |
| | Male | 2.86 (2.22–3.68, p<0.01) | - | 2.39 (1.83–3.13, p<0.01) |
| Age[1] | Under 70 years old | Reference | - | Reference |
| | Greater and equal to 70 years old | 6.50 (4.60–9.18, p<0.01) | - | 5.60 (3.86–8.13, p<0.01) |
| Ethnicity | White | Reference | Reference | Reference |
| | Black | 2.25 (1.23–4.11, p<0.01) | 3.93 (2.14–7.21, p<0.01) | 3.41 (1.79–6.50, p<0.01) |
| | Asian and others | 0.83 (0.44–1.57, p = 0.57) | 1.12 (0.60–2.11, p = 0.72) | 0.84 (0.40–1.76, p = 0.65) |
| BMI[2] | Healthy weight | Reference | Reference | Reference |
| | Underweight | 4.72 (1.46–15.24, p<0.01) | 6.38 (1.97–20.63, p<0.01) | 5.04 (1.55–16.36, p<0.01) |
| | Overweight | 2.05 (1.44–2.92, p<0.01) | 1.59 (1.11–2.27, p = 0.01) | 1.35 (0.94–1.95, p = 0.10) |
| | Obese | 3.75 (2.64–5.32, p<0.01) | 3.17 (2.23–4.50, p<0.01) | 2.16 (1.50–3.12, p<0.01) |
| Drinking frequency | Never | Reference | Reference | Reference |
| | Sometimes | 0.51 (0.34–0.76, p<0.01) | 0.55 (0.37–0.81, p = 0.03) | 0.56 (0.37–0.84, p<0.01) |
| | Weekly | 0.49 (0.34–0.69, p<0.01) | 0.43 (0.30–0.61, p<0.01) | 0.58 (0.40–0.85, p<0.01) |
| | Daily | 0.58 (0.39–0.86, p<0.01) | 0.41 (0.28–0.62, p<0.01) | 0.60 (0.40–0.92, p = 0.02) |
| Smoking status | Non-smoker | Reference | Reference | Reference |
| | Ex-smoker | 2.05 (1.59–2.64, p<0.001) | 1.54 (1.19–1.99, p<0.01) | 1.36 (1.04–1.77, p = 0.03) |
| | Current smoker | 2.13 (1.48–3.06, p<0.001) | 2.16 (1.50–3.12, p<0.01) | 1.53 (1.04–2.25, p = 0.03) |
| Vitamin D status testing time | During non- summertime | Reference | Reference | Reference |
| | During British summer time | 0.97 (0.76–1.23, p = 0.798) | 0.99 (0.78–1.26, p = 0.92) | 1.07 (0.83–1.39, p = 0.59) |
| IMD[3] | Least deprived | Reference | Reference | Reference |
| | 2 deprived | 1.16 (0.73–1.86, p = 0.52) | 1.17 (0.74–1.87, p = 0.50) | 1.00 (0.62–1.61, p = 0.99) |
| | 3 deprived | 1.36 (0.86–2.12, p = 0.19) | 1.40 (0.89–2.19, p = 0.14) | 1.12 (0.71–1.77, p = 0.64) |
| | 4 deprived | 1.78 (1.16–2.72, p<0.01) | 1.94 (1.27–2.97, p<0.01) | 1.36 (0.88–2.12, p = 0.17) |
| | Most deprived | 3.21 (2.17–4.74, p<0.01) | 3.75 (2.53–5.55, p<0.01) | 2.11 (1.39–3.20, p<0.01) |

(*Continued*)

**Table 5.** (Continued)

| | | HR (crude) | HR (adjusted for sex and age) | HR (adjusted for all covariates) |
|---|---|---|---|---|
| Regions | North East | Reference | Reference | Reference |
| | East Midlands | 1.02 (0.64–1.60, p = 0.95) | 0.98 (0.62–1.55, p = 0.95) | 1.16 (0.73–1.84, p = 0.53) |
| | London | 0.46 (0.28–0.75, p<0.01) | 0.51 (0.31–0.82, p<0.01) | 0.46 (0.28–0.77, p<0.01) |
| | North West | 0.81 (0.55–1.20, p = 0.29) | 0.80 (0.54–1.18, p = 0.26) | 0.72 (0.48–1.07, p = 0.11) |
| | South East | 0.18 (0.08–0.41, p<0.01) | 0.18 (0.08–0.41, p<0.01) | 0.22 (0.09–0.55, p<0.01) |
| | South West | 0.32 (0.17–0.61, p<0.01) | 0.34 (0.18–0.65, p<0.01) | 0.44 (0.23–0.85, p = 0.02) |
| | West Midlands | 1.11 (0.74–1.68, p = 0.61) | 1.12 (0.74–1.69, p = 0.60) | 0.98 (0.64–1.50, p = 0.92) |
| | Yorkshire and The Humber | 1.21 (0.85–1.73, p = 0.28) | 1.24 (0.87–1.77, p = 0.24) | 1.29 (0.90–1.85, p = 0.17) |
| Clinically vulnerable to COVID-19[4] | Not vulnerable | Reference | Reference | Reference |
| | Extremely vulnerable | 4.29 (3.40–5.40, p<0.001) | 3.32 (2.63–4.20, p<0.01) | 2.58 (2.02–3.29, p<0.01) |
| Underlying comorbidities[5] | No chronic diseases | Reference | Reference | Reference |
| | Chronic diseases | 4.64 (3.08–7.00, p<0.01) | 2.97 (1.96–4.49, p<0.01) | 1.87 (1.21–2.88, p<0.01) |

[1]. Calculated from participants' year of birth.

[2]. The classification is suggested by NICE guidelines.

[3]. IMD scores were classified by quintile.

[4]. The clinically extremely vulnerable groups were defined by using Public Health England's definition.

[5]. Including hypertension, cardiovascular diseases, and diabetes mellitus

sample size of our study also provides more statistical power than previous studies using single-hospital records. Finally, our analysis showed that some known factors were also associated with COVID-19 hospitalisation and mortality, including male sex, older age, non-white ethnicity, abnormal BMI, cigarette smoking, being more deprived, being clinically vulnerable and having underlying comorbidities. These findings were similar to previous studies using a large electronic health records database [32], implying that our analysis regarding hospitalisation and mortality is valid.

Nevertheless, the study has some limitations. First, the data regarding historical vitamin D status and vitamin D supplementation were collected between 2006 and 2013. The distribution of vitamin D status and supplementation behaviour may be very different now. A previous study among postmenopausal women repeatedly measured vitamin D levels after five years, and the results showed the intraclass correlation coefficient between the two results was only 0.59 (0.54–0.64), which was suboptimal [33]. Another study examined 15,473 people with repeated vitamin D level tests in UK Biobank data, which showed an 84% concordance rate after 4.3 years [30]. However, in our study, the vitamin D levels were measured seven to 15 years ago. The misclassification is likely to be non-differential, which could attenuate our estimates toward the null. In addition, information about self-reported vitamin D supplementation was only available for 54% of participants, which further reduced the statistical power of our analysis and may result in misclassification. Future studies should consider using more

recent data about vitamin D status and more complete vitamin D supplementation information.

Second, the diagnosis of COVID-19 was influenced by testing strategies, which is likely to have led to outcome misclassification. At the early stage of the pandemic in the UK, the testing capacity was limited to people who required inpatient care. Therefore, only participants with relatively severe symptoms were tested, and people who were asymptomatic or had mild symptoms had to stay at home instead of seeking medical care [34]. As the COVID-19 testing capacity increased, more people with mild or no symptoms were able to access testing and classified as cases. Since the COVID-19 testing was not systematic, the outcome of the diagnosis of COVID-19 was misclassified. For future studies, the COVID-19 outcomes should be ascertained systematically.

Third, despite large number of our population, the external validity of UK Biobank is limited. The participants of UK Biobank are not nationally representative, and they are wealthier, older, and more likely to be white and women, which may introduce healthy volunteer bias [35]. However, in our model, we adjusted for demographic covariates, and we also included IMD scores as a proxy for socioeconomic status. Our results regarding exposure and outcomes remain internally valid.

Previous small, single-hospital studies have shown an association between pre-hospitalised vitamin D levels and mortality [36, 37], while other hospital-based studies enrolling more participants have indicated no evidence of such an association [38–40]. We found no evidence that historical vitamin D status was associated with inpatient admission or mortality due to COVID-19, which was similar to another study using UK Biobank data with a shorter follow-up period [30], while we adjusted for more clinical covariates and had a longer follow-up time. However, because the information on vitamin D status from UK Biobank was mainly collected between 2006 and 2010, this may not reflect participants' current vitamin D status. This finding of no association may be biased by misclassification of vitamin D exposure, so results should be interpreted cautiously.

Our study showed inconsistent associations between vitamin D deficiency and diagnosis of COVID-19 during the different follow-up times. During the summertime months, vitamin D deficiency was negatively associated with having a diagnosis of COVID-19 after adjusting for covariates. This result is similar to previous small studies performed early in the pandemic [41, 42] and consistent with other studies using UK Biobank data [28–31]. A possible explanation is that people in the northern hemisphere are less likely to be vitamin D deficient during this period of the year, and during this time, people normally spend less time indoors, which also decreases the risk of being infected with SARS-CoV-2. However, the heterogeneity among the follow-up periods influences the association between vitamin D status and the diagnosis of COVID-19. During British summertime and non-summertime, there were two waves of COVID-19 pandemic caused by different circulating strains [43], and the government's response to COVID-19 also varied. In addition, as previously discussed, the potential misclassification of vitamin D exposure and COVID-19 outcome could have introduced biases, which may lead to inaccurate estimation in the association between vitamin D and COVID-19.

Our study showed no evidence that vitamin D prescription or supplementation was associated with COVID-19 admission or mortality. Previously, a single-hospital study also showed no association between vitamin D supplementation and COVID-19 admission or mortality [38], and a recent meta-analysis also indicated that vitamin D supplements were not associated with COVID-19 mortality reduction [44]. However, the information about vitamin D supplementation of UK Biobank was collected at least 10 years ago, which may not accurately reflect current vitamin D intake. Furthermore, despite adjusting for various clinical covariates, we

still cannot exclude probable residual confounding effects such as confounding by indication. These results should be interpreted carefully in light of these likely biases.

## Conclusion

Our study shows inconsistent associations between vitamin D status and diagnosis of COVID-19, as well as no association between historical vitamin D status and the risk of hospital admission and mortality due to COVID-19. However, these results were limited by the potential for misclassification bias caused by the historical vitamin D status and the changing COVID-19 testing strategies. To precisely investigate the possible role of vitamin D in COVID-19 prevention, more studies using recent vitamin D status data and systematic COVID-19 surveillance will be needed. In addition, the results of an ongoing trial may provide more compelling evidence on the effects of vitamin D supplementation on preventing COVID-19 [45]. Because of the uncertainty of the association between vitamin D deficiency and the risk of COVID-19, there is currently insufficient evidence to support prioritizing vitamin D supplementation or fortification over other preventive strategies for COVID-19, such as mass vaccination programmes.

## Supporting information

**S1 Table. The comparison of inclusion and exclusion participants.**
(DOCX)

**S2 Table. The association between receiving vitamin D tests during British summer time and serum vitamin D status.**
(DOCX)

**S3 Table. The association between vitamin D prescription and Covid-19 diagnosis.**
(DOCX)

**S4 Table. The association between vitamin D prescription and hospitalization due to Covid-19.**
(DOCX)

**S5 Table. The association between vitamin D prescription and mortality due to Covid-19.**
(DOCX)

**S6 Table. The association between vitamin D supplementation and Covid-19 diagnosis.**
(DOCX)

**S7 Table. The association between vitamin D supplementation and hospitalization due to Covid-19.**
(DOCX)

**S8 Table. The association between vitamin D supplementation and mortality due to Covid-19.**
(DOCX)

**S9 Table. The association between vitamin D status and laboratory-confirmed Covid-19 diagnosis.**
(DOCX)

**S10 Table. The association between vitamin D status and hospital admission among patients with Covid-19 diagnosis.**
(DOCX)

**S11 Table. The association between vitamin D status and mortality among patients with Covid-19 diagnosis.**
(DOCX)

**S12 Table. The data recording dates and data access dates.**
(DOCX)

**S1 Fig. The distribution of COVID-19 outcomes.**
(DOCX)

**S2 Fig. The log(-log(survival)) plot of Cox models for the primary and secondary outcomes.**
(DOCX)

## Acknowledgments

We thank Dr Helen McDonald and Dr Jemma Walker and Dr Christopher Rentsch for assistance with clinical code lists and prescription data. We also thank UK Biobank for approving our project and providing research data (project ID:51265).

## Author Contributions

**Conceptualization:** Liang-Yu Lin, Amy Mulick, Rohini Mathur, Charlotte Warren-Gash, Sinéad M. Langan.

**Formal analysis:** Liang-Yu Lin.

**Investigation:** Liang-Yu Lin.

**Methodology:** Liang-Yu Lin, Amy Mulick, Rohini Mathur, Charlotte Warren-Gash, Sinéad M. Langan.

**Project administration:** Liang-Yu Lin.

**Software:** Liang-Yu Lin, Amy Mulick.

**Supervision:** Amy Mulick, Rohini Mathur, Liam Smeeth, Charlotte Warren-Gash, Sinéad M. Langan.

**Visualization:** Liang-Yu Lin.

**Writing – original draft:** Liang-Yu Lin.

**Writing – review & editing:** Liang-Yu Lin, Amy Mulick, Rohini Mathur, Liam Smeeth, Charlotte Warren-Gash, Sinéad M. Langan.

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
