## [Decision Letter · Decision Letter 0]

28 Dec 2021

PONE-D-21-15035

The association between vitamin D status and COVID-19 in England: a cohort study using UK Biobank

PLOS ONE

Dear Dr. Lin,

Thank you for submitting your manuscript to PLOS ONE. After careful consideration, we feel that it has merit but does not fully meet PLOS ONE’s publication criteria as it currently stands. Therefore, we invite you to submit a revised version of the manuscript that addresses the points raised during the review process.

We look forward to receiving your revised manuscript.

Kind regards,

Sreeram V. Ramagopalan

Academic Editor

PLOS ONE

2. Thank you for providing the date(s) when patient medical information was initially recorded. Please also include the date(s) on which your research team accessed the databases/records to obtain the retrospective data used in your study.

3. In your ethics statement in the Methods section and in the online submission form, please provide additional information about the data used in your retrospective study. Specifically, please ensure that you have discussed whether all patient data were fully anonymized before you accessed them and/or whether the IRB or ethics committee waived the requirement for informed consent. If patients provided informed written consent to have data from their medical records used in research, please include this information.

4. Please include your actual numerical p-values in Tables 3-5 and supplementary tables 2-11

5.  Thank you for stating the following in the Funding Section of your manuscript:

“LYL is funded by the scholarship of government sponsorship for overseas study by the Ministry of Education Republic of China (Taiwan). RM is funded by a Sir Henry Wellcome Postdoctoral Fellowship (201375/Z/16/Z). CWG is supported by a Wellcome Intermediate Clinical Fellowship (201440/Z/16/Z). SML is funded by a Wellcome Senior Clinical Fellowship in Science (205039/Z/16/Z).”

We note that you have provided additional information within the Funding Section that is not currently declared in your Funding Statement. Please note that funding information should not appear in other areas of your manuscript. We will only publish funding information present in the Funding Statement section of the online submission form.

 “The authors received no specific funding for this work.”

Reviewers' comments:

Reviewer's Responses to Questions

**Comments to the Author**

1. Is the manuscript technically sound, and do the data support the conclusions?

Reviewer #1: No

2. Has the statistical analysis been performed appropriately and rigorously? 

Reviewer #1: No

3. Have the authors made all data underlying the findings in their manuscript fully available?

Reviewer #1: Yes

4. Is the manuscript presented in an intelligible fashion and written in standard English?

Reviewer #1: No

5. Review Comments to the Author

Reviewer #1: This could be an important paper that is worth of publication but has multiple major flaws that must be addressed before it can be seriously considered for publication,

1. The authors are unclear about the primary outcome, which I think is a diagnosis of COVID 19.

2. There is insufficient distinction made between " not statistically significant" and underpowered for the outcome. The hospitalization and mortality outcomes are underpowered for likely and clinically important effect sizes

3. There is some attention given to the historical nature of the vitamin D levels but the import of those lags and likely interval treatments and changes is not seriously enough appreciated in the conclusions of the paper nor are there any efforts to discuss whether they can be usefully analyzed with this data.

4. The finding of significant association of vitamin D deficiency with covid-19 positivity controlling for covariates in the winter seems to be a major finding of the paper and is neglected by the authors. This is especially important because there are strong reasons to believe that summer sun exposure confounds vitamin D levels for COVID risk in the summer

5. I do not understand why results unadjusted for known risk factors such as gender are presented. This seems to muddy the presentation

6. The term non-british summertime is awkward. It sounds like summertime not in Britain.

7. The conclusion is disappointing in suggesting that this work suggests that vitamin D should not be prioritized over vaccination. That is clear and NOT the question. The question is is there evidence vitamin D might play a protective role in COVID-19, and might it be worth studying or trying even outside of studies given the low risks. This is what the authors should be commenting on.

6. PLOS authors have the option to publish the peer review history of their article (what does this mean?). If published, this will include your full peer review and any attached files.

Reviewer #1: No

---

## [Author Response · Author response to Decision Letter 0]

7 Apr 2022

Response to reviewer

Reviewer #1: This could be an important paper that is worth of publication but has multiple major flaws that must be addressed before it can be seriously considered for publication,

We thank the reviewer for the helpful comments. We noticed some coding errors in our analysis, which previously over diagnosed people who were extremely vulnerable to COVID-19. These errors now have been corrected as follows (page 16, line 257): 

Table 2. The distribution of demographic characteristics by vitamin D status

We have addressed other comments as below:

1. The authors are unclear about the primary outcome, which I think is a diagnosis of COVID 19.

Thank you for the comment. The reviewer is correct. We have revised the words in the methods sections as follows to clarify (page 8, line 116–128): 

Primary outcome: diagnosis of COVID-19 

The primary outcome of our study was the diagnosis of COVID-19, which was defined through laboratory testing or by clinical diagnosis of COVID-19. The laboratory tests for SARS-CoV-2 infection were performed using PCR, which was performed by the NHS (Pillar 1) or commercial partners (Pillar 2) (18, 19). These testing results were reported to Public Health England and automatically imported into UK Biobank weekly (20). Clinically diagnosed COVID-19 was defined as participants diagnosed with COVID-19 codes in their electronic health records, either in primary care or inpatient care, or on the death certificate. We used existing code lists in CTV3 codes, SNOMED-CT and ICD-10 to identify the diagnosis of COVID-19.

2. There is insufficient distinction made between " not statistically significant" and underpowered for the outcome. The hospitalization and mortality outcomes are underpowered for likely and clinically important effect sizes

Thank you for this comment. We agree with the referee that the wide confidence intervals and large p-values for the hospitalization and mortality outcomes indicate that our results were imprecise, which implies they were underpowered to detect the effects we presented [7]. With our large SEs, we may have been able to detect larger hospitalization/mortality effects if they existed in our data, but they did not. We have clarified in the text that the confidence intervals were consistent with other clinically important effects in increasing the risk of COVID-19 as well as no effect, none of which can be ruled out by our analysis. We have address this in the discussion (page 39, line 392–397):

“… Finally, our analysis showed that some known factors were also associated with COVID-19 hospitalisation and mortality, including male sex, older age, non-white ethnicity, abnormal BMI, cigarette smoking, being more deprived, being clinically vulnerable and having underlying comorbidities. These findings were similar to previous studies using a large electronic health records database [32], implying that our analysis regarding hospitalisation and mortality is valid.”

3. There is some attention given to the historical nature of the vitamin D levels but the import of those lags and likely interval treatments and changes is not seriously enough appreciated in the conclusions of the paper nor are there any efforts to discuss whether they can be usefully analyzed with this data.

The referee is right to point out the importance of discussing the effects of using historical vitamin D measurement. We have now addressed this point in the discussion as follows (page 40, line 400–407): 

“A previous study among postmenopausal women repeatedly measured vitamin D levels after five years, and the results showed the intraclass correlation coefficient between the two results was only 0.59 (0.54–0.64), which was suboptimal [8]. Another study examined 15,473 people with repeated vitamin D level tests in UK Biobank data, which showed an 84% concordance rate after 4.3 years [9]. However, in our study, the vitamin D levels were measured seven to 15 years ago, which introduces a possible misclassification bias of exposure. The misclassification is likely to be non-differential, which could attenuate our estimates toward the null.”

In addition, we have also now addressed this limitation in a revised conclusion as follows (page 42, line 462–474): 

“Our study shows inconsistent associations between vitamin D status and diagnosis of COVID-19, as well as no association between historical vitamin D status and the risk of hospital admission and mortality due to COVID-19. However, these results were limited by the potential for misclassification bias caused by the historical vitamin D status and the changing COVID-19 testing strategies. To precisely investigate the possible role of vitamin D in COVID-19 prevention, more studies using recent vitamin D status data and systematic COVID-19 surveillance will be needed. In addition, the results of the ongoing trial may provide more compelling evidence on the effects of vitamin D supplementation on preventing COVID-19 [10]. Because of the uncertainty of the association between vitamin D deficiency and the risk of COVID-19, there is currently insufficient evidence to support prioritizing vitamin D supplementation or fortification over other preventive strategies for COVID-19 such as mass vaccination programmes.”

4. The finding of significant association of vitamin D deficiency with covid-19 positivity controlling for covariates in the winter seems to be a major finding of the paper and is neglected by the authors. This is especially important because there are strong reasons to believe that summer sun exposure confounds vitamin D levels for COVID risk in the summer

Although we agree with the referee that there was some evidence of the association between vitamin D deficiency and COVID-19 during non-summertime, we would like to address the significant heterogeneity between follow-up periods. First, the stratification of follow-up period not only showed the potential difference in the sun exposure, but also reflected different properties of COVID-19 pandemic, such as different waves of the outbreak caused by more virulent variant strains, or different strategies for COVID-19 response. Because of these heterogeneity between two follow-up times, we would like to not over focus on a marginal significant result during non-summertime. Second, however, as mentioned in the previous paragraphs, the misclassification bias caused by the historical vitamin D levels and the testing strategies should also be considered. Because of these limitations, we cautiously interpreted our results as an inconsistent association between vitamin D status and diagnosis of COVID-19. We addressed these two points in the discussion as follows (page 41, line 444–451): 

“However, the heterogeneity among the follow-up periods influences the association between vitamin D status and the diagnosis of COVID-19. During British summertime and non-summertime, there were two waves of COVID-19 pandemic caused by different circulating strains [11], and the government’s response to COVID-19 also varied. In addition, as previously discussed, the potential misclassification of vitamin D exposure and COVID-19 outcome also could have introduced biases, which may lead to inaccurate estimation in the association between vitamin D and COVID-19.“

5. I do not understand why results unadjusted for known risk factors such as gender are presented. This seems to muddy the presentation

We understand the reviewer’s concern. However, as can be seen in our tables, we considered that presenting crude, minimally adjusted, and fully adjusted estimates aids interpretation because it demonstrates both the extent of confounding, and the covariates responsible. Presenting all potential confounders in models is useful to demonstrate the effect of confounding. 

6. The term non-British summertime is awkward. It sounds like summertime not in Britain.

We agree with the referee that using "non-British summertime" is misleading. Therefore, we've changed the term into "non-summertime" in the main text and figures. 

7. The conclusion is disappointing in suggesting that this work suggests that vitamin D should not be prioritized over vaccination. That is clear and NOT the question. The question is there evidence vitamin D might play a protective role in COVID-19, and might it be worth studying or trying even outside of studies given the low risks. This is what the authors should be commenting on.

We agree with the referee that the study's aim, implications, and conclusion should be clearer. Therefore, we have revised our introduction and conclusion. We've revised the last paragraph of our introduction as follows (page 5, line 86–91): 

“Despite this potential, the association between vitamin D and COVID-19 is still unclear. If vitamin D deficiency is associated with COVID-19, vitamin D supplementation may be a potential public health intervention. Consequently, we aimed to conduct a historical cohort study using UK Biobank dataset and linked electronic health records to better understand the association between serum vitamin D status, vitamin D supplementation and diagnosis of COVID-19 and outcomes.”

We also revised the conclusion as follows (page 42, line 462–474): 

“Our study shows inconsistent associations between vitamin D status and diagnosis of COVID-19, as well as no association between historical vitamin D status and the risk of hospital admission and mortality due to COVID-19. However, these results were limited by the potential for misclassification bias caused by the historical vitamin D status and the changing COVID-19 testing strategies. To precisely investigate the possible role of vitamin D in COVID-19 prevention, more studies using recent vitamin D status data and systematic COVID-19 surveillance will be needed. In addition, the results of an ongoing trial may provide more compelling evidence on the effects of vitamin D supplementation on preventing COVID-19 [10]. Because of the uncertainty of the association between vitamin D deficiency and the risk of COVID-19, there is currently insufficient evidence to support prioritizing vitamin D supplementation or fortification over other preventive strategies for COVID-19 such as mass vaccination programmes.”

References 

1. UK Biobank. Summary de-identification protocol 2021 [updated 6 July 2021; cited 2021 4 December]. 2:[Available from: https://www.ukbiobank.ac.uk/media/5bvp0vqw/de-identification-protocol.pdf.

2. UK Biobank. UK Biobank ethics and governance framework 2007 [updated October 2007; cited 2022 28 Feb]. 3.0:[Available from: https://www.ukbiobank.ac.uk/media/0xsbmfmw/egf.pdf.

3. UK Biobank. UK Biobank - COVID-19 Data Release FAQs 2020 [cited 2020 26, May, 2020]. Available from: https://www.ukbiobank.ac.uk/wp-content/uploads/2020/05/ACCESS_066V1.1.A.COVID-19-FAQs_v1.1_DRAFT.docxAMENDEDBH.pdf.

4. World Medical A. World Medical Association Declaration of Helsinki: ethical principles for medical research involving human subjects. JAMA. 2013;310(20):2191-4. Epub 2013/10/22. doi: 10.1001/jama.2013.281053. PubMed PMID: 24141714.

5. Sudlow C, Gallacher J, Allen N, Beral V, Burton P, Danesh J, et al. UK biobank: an open access resource for identifying the causes of a wide range of complex diseases of middle and old age. PLoS Med. 2015;12(3):e1001779. Epub 2015/04/01. doi: 10.1371/journal.pmed.1001779. PubMed PMID: 25826379; PubMed Central PMCID: PMCPMC4380465.

6. UK Biobank. Consent Form: UK Biobank.

7. Althouse AD. Post Hoc Power: Not Empowering, Just Misleading. J Surg Res. 2021;259:A3-a6. Epub 20200816. doi: 10.1016/j.jss.2019.10.049. PubMed PMID: 32814615.

8. Meng JE, Hovey KM, Wactawski-Wende J, Andrews CA, LaMonte MJ, Horst RL, et al. Intraindividual Variation in Plasma 25-Hydroxyvitamin D Measures 5 Years Apart among Postmenopausal Women. Cancer Epidemiology Biomarkers & Prevention. 2012;21(6):916. doi: 10.1158/1055-9965.EPI-12-0026.

9. Hastie CE, Pell JP, Sattar N. Vitamin D and COVID-19 infection and mortality in UK Biobank. Eur J Nutr. 2021;60(1):545-8. Epub 2020/08/26. doi: 10.1007/s00394-020-02372-4. PubMed PMID: 32851419.

10. Trial of vitamin D to reduce risk and severity of COVID-19 and other acute respiratory infections. Identifier: NCT04579640. 2020.

11. Office for National Statistics. Coronavirus (COVID-19) Infection Survey technical article: waves and lags of COVID-19 in England, June 2021 2021 [updated 29 June 2021; cited 2022 4 March]. Available from: https://www.ons.gov.uk/peoplepopulationandcommunity/healthandsocialcare/conditionsanddiseases/articles/coronaviruscovid19infectionsurveytechnicalarticle/wavesandlagsofcovid19inenglandjune2021.

---

## [Editor Report · Decision Letter 1]

16 May 2022

The association between vitamin D status and COVID-19 in England: a cohort study using UK Biobank

PONE-D-21-15035R1

Dear Dr. Lin,

We’re pleased to inform you that your manuscript has been judged scientifically suitable for publication and will be formally accepted for publication once it meets all outstanding technical requirements.

Kind regards,

Sreeram V. Ramagopalan

Academic Editor

PLOS ONE
---

## [Editor Report · Acceptance letter]

26 May 2022

PONE-D-21-15035R1 

The association between vitamin D status and COVID-19 in England: a cohort study using UK Biobank 

Dear Dr. Lin:

I'm pleased to inform you that your manuscript has been deemed suitable for publication in PLOS ONE. Congratulations! Your manuscript is now with our production department. 

Kind regards, 

on behalf of

Dr. Sreeram V. Ramagopalan 

Academic Editor

PLOS ONE